# LEVERAGING LLM EMBEDDINGS FOR CROSS DATASET LABEL ALIGNMENT AND ZERO SHOT MUSIC EMOTION PREDICTION

## ABSTRACT

In this work, we present a novel method for music emotion recognition that leverages Large Language Model (LLM) embeddings for label alignment across multiple datasets and zero-shot prediction on novel categories. First, we compute LLM embeddings for emotion labels and apply non-parametric clustering to group similar labels, across multiple datasets containing disjoint labels. We use these cluster centers to map music features (MERT) to the LLM embedding space. To further enhance the model, we introduce alignment regularization that enables dissociation of MERT embeddings from different clusters. This further enhances the model's ability to better adaptation to unseen datasets. We demonstrate the effectiveness of our approach by performing zero-shot inference on a new dataset, showcasing its ability to generalize to unseen labels without additional training.

## 1 INTRODUCTION

The task of automatic music emotion recognition has been a long-standing challenge in the field of music information retrieval (Yang & Chen, 2012; Kim et al., 2010; Kang & Herremans, 2024). Accurately predicting the emotional impact of music has numerous valuable applications, ranging from improving music streaming recommendations to providing more effective tools for music therapists. By understanding the emotional responses evoked by music, we can improve the user experience of music listening platforms, tailoring recommendations to individual preferences and emotional needs. Furthermore, this knowledge can benefit music therapists, enabling them to select and apply music more effectively in their treatments, ultimately leading to better outcomes for their patients (Agres et al., 2021).

The predominant approach in this field has been to model emotions using Russell's two dimensional valence-arousal space (Russell, 1980). However, this representation fails to be interpreted by human (Eerola & Vuoskoski, 2011). to address this limitation, researchers have explored the use of comprehensive categorical emotional models, such as the Geneva Emotional Music Scale (Aljanaki et al., 2014), which encompasses a broad range of discrete emotional attributes. More recent datasets (Bogdanov et al., 2019; Turnbull et al., 2007) introduce new label categories, sometimes in the form of free tags, that aim to capture the multifaceted nature of human emotions. Nonetheless, a key challenge in this area is the lack of a unified emotional taxonomy. Different datasets often employ disparate and often incompatible sets of emotional labels, posing a significant obstacle. The heterogeneity of emotion label taxonomies across datasets hinders the ability to effectively compare and combine findings across studies. This impedes our comprehensive understanding of the nuanced emotional responses to music. The challenge of aligning these disparate emotion label taxonomies limits our capacity to develop more comprehensive and robust music emotion recognition models. Traditionally, the majority of studies have focused on training and testing with a single dataset. However, recent advances in large language models have resulted in powerful general-purpose text encoders (Reimers, 2019) that can be leveraged to effectively align emotions across diverse datasets. The ability to align emotion labels across disparate datasets is a crucial step towards developing more comprehensive and robust music emotion recognition models that can generalise to a wider range of emotional experiences.

In this work, we present a novel methodology that leverages the powerful representational capabilities of Large Language Model embeddings to enable effective cross-dataset label alignment and facilitate zero-shot inference on new datasets with previously unseen emotion labels. The core of our approach involves computing LLM embeddings for the emotion labels across multiple datasets, and then applying non-parametric clustering to group semantically similar labels together. These cluster centres serve as anchor points that allow us to map the MERT features (Li et al., 2023) extracted from music wav files into a common label embedding space, effectively aligning the disparate emotion taxonomies present across datasets. To further enhance the model's performance and ensure it captures the underlying semantic relationships between emotions beyond training data, we introduce an alignment regularization. This regularization encourages music features from differ clusters in the label embedding space to dissociate from each other. This, in turn help the model to generalize better to unseen data and labels. To evaluate the efficacy of our proposed approach, we conduct extensive experiments on three benchmark datasets for music emotion recognition. The results demonstrate substantial improvements in the model's ability to perform zero-shot prediction on a new dataset containing previously unseen emotion labels, showcasing the strong generalization capabilities of our method. This is a crucial step towards developing more robust and comprehensive music emotion recognition models that can be applied to a wider range of emotional experiences beyond the singular datasets used during training.

To sum up, the key contributions of this work are:

- This work is the first to leverage the capabilities of Large Language Model embeddings to enable robust cross-dataset label alignment for the domain of music emotion recognition. By computing LLM embeddings for emotion labels across datasets and applying non-parametric clustering, we are able to establish a common embedding space that allows us to align disparate emotion taxonomies.

- We introduce an alignment regularization that further enhances the model's ability to dissociate music features from distinct emotions.

- We demonstrate the ability of our approach to perform zero-shot inference on new datasets with previously unseen emotion labels. This showcases the model's strong generalization capabilities and its potential to be applied to a wider range of emotional experiences.

The rest of the paper is organized as follows. Related works in music emotion recognition and cross-dataset transfer learning are discussed in Section 2. We describe our proposed label alignment and emotion prediction framework in detail in Section 3. Section 4 elaborates on our experimental setup, followed by a comprehensive discussion of the results. Finally we conclude the paper and outline future research directions in Section 5.

## 2 RELATED WORK

The connection between music and emotions has long been a subject of study, dating back to (Leonard, 1956) and (Hevner, 1935), who explored how different musical elements evoke specific emotional responses. Over the decades, various frameworks have been proposed to represent emotions in music, ranging from categorical models (e.g., happy, sad, angry) to continuous dimensions like valence and arousal. Valence-arousal models (Russell, 1980) and more sophisticated systems such as the Geneva Emotional Music Scale (Zentner et al., 2008) have become prominent in recent studies of music emotion recognition (MER).

Despite these advances, current models often struggle to achieve robust generalization, particularly across diverse datasets. Existing approaches have attempted to improve performance by incorporating advanced encoders like MERT (Li et al., 2023), leveraging multi-modal data (e.g., lyrics, MIDI, or video), or introducing personalized models that adapt to listener-specific responses (Chua et al., 2022; Koh et al., 2022; Sams & Zahra, 2023). However, these methods primarily explore single dataset approach, i.e., training and testing done on a single dataset. Compared to other fields, such as image recognition or natural language processing, datasets for MER are considerably smaller and more fragmented, making it difficult to develop models that generalize across genres and emotional taxonomies. Datasets such as MTG-Jamendo (Bogdanov et al., 2019), and smaller, domain-specific collections like CAL500 (Turnbull et al., 2007) and Emotify (Aljanaki et al., 2016) all use different emotion representation models, focus on distinct musical genres, and are thus often incompatible.

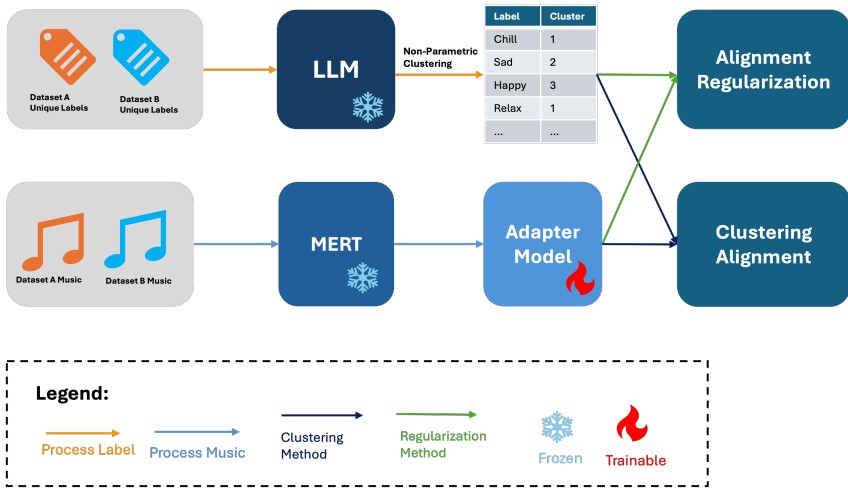

Figure 1: Overview of our approach.

This discrepancy in representation and dataset size presents a major challenge to building robust models that can generalize across various emotion taxonomies. Addressing this limitation requires innovative solutions that can overcome the dataset heterogeneity. One promising direction is zero-shot learning, which has shown success in fields such as computer vision (Xian et al., 2018) and natural language processing (Brown, 2020). Zero-shot learning models generalize to novel classes or tasks without requiring direct training data for every label. In the context of music emotion recognition, zero-shot learning enables models to predict emotions in datasets with unseen labels or emotion taxonomies, potentially overcoming the generalization bottleneck caused by small, disjoint datasets. Our work builds on these ideas by introducing a novel approach for emotion prediction across music datasets with disjoint label sets. By leveraging large language models (LLMs) to align emotion labels through semantic embeddings and clustering, we bridge the gap between datasets, enabling cross-dataset generalization and zero-shot performance. The next section provides details of the building blocks of our approach.

## 3 PRESENT WORK

Our approach leverages the power of Large Language Model (LLM) embeddings to align emotion labels across multiple music emotion datasets and to enable zero-shot inference on previously unseen emotion labels. An overview is shown in Figure 1. This section describes the key components of our methodology, including the label embedding procedure, non-parametric clustering, and the mapping of MERT features (i.e., encoded audio) (Li et al., 2023) to the label embedding space.

### 3.1 EMOTION LABEL EMBEDDING

For each dataset, we obtain an embedding for each emotion label using a pre-trained LLM. Let $\mathcal{L}_d = \{l_1, l_2, \ldots, l_{n_d}\}$ represent the set of emotion labels in dataset $d$, where $n_d$ is the number of labels in dataset $d$. We compute an embedding for each label $l_i \in \mathcal{L}_d$ using the LLM, which produces a vector representation $\mathbf{e}_{l_i} \in \mathbb{R}^m$, where $m$ is the dimension of the LLM's embedding space:

$$\mathbf{e}_{l_i} = \text{LLM}(l_i) \tag{1}$$

This embedding process is performed independently for each dataset. The advantage of using LLM embeddings is that the pre-trained model captures rich semantic relationships between words (emotion labels), which allows for natural grouping of labels even when they differ slightly across datasets. For example, the labels "happy" and "joyful" are likely to have similar embeddings despite being distinct.

## 3.2 CLUSTERING OF EMOTION EMBEDDINGS

Once we have the LLM embeddings for the emotion labels from multiple datasets, the next step is to group semantically similar labels. Instead of relying on a predefined number of clusters, we utilize the Mean Shift clustering algorithm (Cheng, 1995), a non-parametric clustering technique that automatically determines the number of clusters based on the density of points in the embedding space.

Let $\mathcal{E} = \{\mathbf{e}_{l_i}\}_{i=1}^{N}$ represent the set of all emotion label embeddings, where $N$ is the total number of unique labels across all datasets. The Mean Shift algorithm identifies clusters by shifting each point towards the mode (the densest region of points) iteratively.

The algorithm converges when the embedding positions no longer change significantly, resulting in a set of cluster centers $\mathcal{C} = \{\mathbf{c}_1, \mathbf{c}_2, \ldots, \mathbf{c}_K\}$.

## 3.3 MAPPING MERT FEATURES TO THE LABEL EMBEDDING SPACE

Features extracted with the Music undERstanding model with large-scale self-supervised Training (MERT) (Li et al., 2023) have shown to achieve state-of-the-art performance in various music understanding tasks. MERT features can be seen as a highly expressive and generalizable feature representation for music. We leverage these MERT features as the input representation for music emotion recognition. These features, however, do not directly correspond to the semantic meaning of emotion labels. To bridge this gap, we propose a mapping from the MERT feature space to the LLM emotion embedding space. We employ an attention-based mechanism to map the MERT features into the label embedding space wherein the emotion label embeddings reside. The model ingests MERT features extracted from music and outputs corresponding embeddings that are aligned with the LLM-generated emotion label embeddings. Let $\mathbf{x}_i \in \mathbb{R}^d$ be the MERT feature matrix for the $i$-th music sample, where $d$ is the feature dimension. The model uses self-attention mechanisms to process this input, applying two encoder layers and projecting it into the label embedding space. The model can be mathematically described as follows:

### 3.3.1 INPUT PROJECTION

We select the `3rd`, `6th`, `9th`, and `12th` layers from MERT and concatenate them to linearly project to a vector of dimension $d_{\text{model}}$. These specific layers were chosen based on empirical experiments (see Appendix A1), which demonstrated that they provide optimal performance.

$$\mathbf{z}_i = \text{Linear}_{\text{proj}}\left(\text{Concat}(\mathbf{x}_i^{(3)}, \mathbf{x}_i^{(6)}, \mathbf{x}_i^{(9)}, \mathbf{x}_i^{(12)})\right), \quad \mathbf{z}_i \in \mathbb{R}^{d_{\text{model}}} \tag{2}$$

### 3.3.2 SELF-ATTENTION

The projected vector $\mathbf{z}_i$ is passed through multiple self-attention layers, where each layer applies multi-head attention followed by a feedforward neural network and layer normalization. The output of the $l$-th attention block is:

$$\mathbf{a}_i^{(l+1)} = \text{LayerNorm}\left(\text{SelfAttention}\left(\mathbf{a}_i^{(l)}\right) + \mathbf{a}_i^{(l)}\right), \quad l = 1, 2, \ldots, L \tag{3}$$

where $\mathbf{a}_i^{(0)} = \mathbf{z}_i$.

### 3.3.3 OUTPUT PROJECTION

The final output from the last attention block is linearly projected to the output size $m$, where $m$ is the dimension of the shared LLM embedding space:

$$\mathbf{f}_\theta(\mathbf{x}_i) = \text{Linear}_{\text{output}}(\mathbf{a}_i^{(L)}) \tag{4}$$

Thus, $\mathbf{f}_\theta(\mathbf{x}_i) \in \mathbb{R}^m$ represents the projected embedding of the MERT features into the shared emotion label embedding space.

### 3.3.4 ALIGNMENT LOSS

To facilitate alignment, we employ a *triplet loss* based alignment loss that ensures the model learns to bring MERT embeddings closer to the LLM embeddings of their true emotion labels, while pushing them apart from embeddings of incorrect emotion categories. In this formulation:

- The **anchor** $\mathbf{f}_\theta(\mathbf{x}_i)$ represents the MERT feature of the audio sample $\mathbf{x}_i$ passed through the model.
- The **positive** $\mathbf{e}_{\text{LLM}}(y_i)$ is the LLM embedding of the true label $y_i$ for the sample.
- The **negative** $\mathbf{e}_{\text{LLM}}(y_k)$ is the LLM embedding of an incorrect or different emotion label $y_k$ (where $y_k \neq y_i$).

The triplet loss is defined based on cosine similarity:

$$\mathcal{L}_{\text{align}} = \frac{1}{N} \sum_{i=1}^{N} \max\left(0, \cos\left(\mathbf{f}_\theta(\mathbf{x}_i), \mathbf{e}_{\text{LLM}}(y_i)\right) - \cos\left(\mathbf{f}_\theta(\mathbf{x}_i), \mathbf{e}_{\text{LLM}}(y_k)\right) + \text{margin}\right) \quad (5)$$

Where:

- $\cos(a, b)$ is the cosine similarity between vectors $a$ and $b$,
- margin is a hyperparameter that defines the minimum desired separation between positive and negative pairs,
- $\mathbf{f}_\theta(\mathbf{x}_i)$ is the model output for the MERT feature $\mathbf{x}_i$,
- $\mathbf{e}_{\text{LLM}}(y_i)$ and $\mathbf{e}_{\text{LLM}}(y_k)$ are the LLM embeddings for the true and incorrect labels, respectively.

This loss function encourages the model to map MERT features closer to their corresponding emotion label embeddings in the label embedding space while ensuring that they are distinct from embeddings of incorrect labels. Ultimately, this mapping procedure allows us to project music representations into the same semantic space as the emotion labels, facilitating more meaningful and interpretable emotion recognition. By aligning the music features and emotion labels in a label embedding space, we can better capture the nuanced relationships between musical characteristics and emotional responses.

### 3.4 ALIGNMENT REGULARIZATION

To further improve the alignment between the MERT feature representations and the emotion label embeddings, we introduce an alignment regularization term. This regularization encourages the model to map MERT features with semantically similar emotion labels to nearby positions in the label embedding space. By minimizing the distance between the embeddings of MERT features corresponding to the same or closely related emotion labels, the model is incentivised to position these representations in close proximity within the label embedding space. The alignment regularization is formulated as follows:

Let $\mathcal{C}_k$ represent the set of samples in cluster $k$, and $\mathbf{x}_i \in \mathcal{C}_{k_l}$, $\mathbf{x}_j \in \mathcal{C}_{k_m}$ be two MERT feature vectors whose corresponding emotion labels belong to the different clusters, i.e., $\mathcal{C}_{k_l} \neq \mathcal{C}_{k_m}$. The alignment regularization term maximizes the distance between their embeddings $\mathbf{f}_\theta(\mathbf{x}_i)$ and $\mathbf{f}_\theta(\mathbf{x}_j)$ in the label embedding space. We define the alignment regularization loss as:

$$\mathcal{L}_{\text{reg}} = \frac{1}{K} \sum_{(x_i, x_j)} 1 - \mathcal{D}(\mathbf{f}_\theta(\mathbf{x}_i), \mathbf{f}_\theta(\mathbf{x}_j)) \quad (6)$$

Where:

- $\mathbf{f}_\theta(\mathbf{x}_i)$ and $\mathbf{f}_\theta(\mathbf{x}_j)$ are the model outputs for the MERT features $\mathbf{x}_i$ and $\mathbf{x}_j$,
- $\mathcal{D}$ represents the cosine distance,
- The sum is taken over all $\mathbf{x}_i \in \mathcal{C}_{k_l}$, $\mathbf{x}_j \in \mathcal{C}_{k_m}$ such that $\mathcal{C}_{k_l} \neq \mathcal{C}_{k_m}$. $K$ is the number of such pairs present in the dataset.

By minimizing $\mathcal{L}_{\text{align}}$, the model encourages MERT features from different labels to dissociate further with each other. The final objective function combines the alignment loss and the regularization term, with a hyperparameter $\lambda$ controlling the trade-off between these two components:

$$\mathcal{L} = \mathcal{L}_{\text{align}} + \lambda \mathcal{L}_{\text{reg}} \qquad (7)$$

The network is trained to minimize $\mathcal{L}$ and we posit this regularization helps the model to generalise to unseen emotion labels and enables it to perform zero shot classification on new datasets. To test this, we set up the following zero shot evaluation scenario described in the next subsection.

### 3.5 Zero Shot Classification

We perform zero-shot inference on a new dataset containing disjoint set of labels, some of which can be previously unseen. Given a new emotion label $l_{\text{new}}$, we compute its embedding $\mathbf{e}_{l_{\text{new}}}$ using the LLM:

$$\mathbf{e}_{l_{\text{new}}} = \text{LLM}(l_{\text{new}}) \qquad (8)$$

To predict the top $k$ emotions for a given music sample, we project the MERT features of the sample into the LLM embedding space using the trained network $f_\theta$. The predicted emotions $\hat{y}$ is given by the following equation:

$$\hat{y} = \arg \min_i {}_k \mathcal{D}(f_\theta(\mathbf{x}), \mathbf{y}_i) \qquad (9)$$

Where $(D)$ is the cosine distance, $k$ is number of top predictions to select. In our experiments we fix $k$ to 2/3/4 depending on dataset.

In the next section, we detail our empirical setup and demonstrate the effectiveness of our approach.

## 4 Experiments and Results

To demonstrate the efficacy of our approach, we evaluate on three distinct emotion recognition datasets. The first subsection provides a concise overview of the datasets. This is then followed by the experimental setup, and finally, we present the results accompanied by a discussion.

### 4.1 Datasets

For the purpose of evaluating our music emotion recognition approach, we have selected three prominent datasets in this research domain: the MTG-Jamendo Dataset (Bogdanov et al., 2019), the Computer Audition Lab 500 dataset (Turnbull et al., 2007), and the the Emotify Dataset (Aljanaki et al., 2016).

The **MTG-Jamendo dataset** is an openly available resource for the task of automatic music tagging. It was constructed using music content hosted on the Jamendo platform, which is licensed under Creative Commons, and incorporating tags provided by the content contributors. This dataset encompasses a vast collection of over 55,000 full-length audio recordings annotated with 56 relevant emotion tags.

The **Computer Audition Lab 500 (CAL500) dataset** is a widely utilised dataset in music emotion recognition research. It consists of 500 popular Western songs, each annotated with a standard set of 17 emotion labels by at least 3 human annotators.

Lastly, the **Emotify dataset** is another prominent open dataset for music emotion recognition. It contains 400 music excerpts annotated with 9 emotional categories of the Geneva Emotional Music Scale model, obtained through a crowdsourcing game.

Both the CAL500 and Emotify datasets feature annotations from multiple users. For the CAL500 dataset, we consider a label as true if its average score is greater than 3 on a scale of 1 to 5. In case none of the emotion score is greater than 3, we select the highest scored emotion. For the Emotify dataset, we fix the selection process by choosing the top 3 rated labels as the true labels. These

processed labels are made available online to allow future benchmarking with our work[1]. It is worth noting that the MTG-Jamendo dataset already incorporates multiple emotion tags, and thus all our datasets are inherently multi-label in nature. Table 1 contains a brief overview of all the datasets.

| Dataset | Number of labels | Training data | Validation data | Test data | Average duration (s) |
|---|---|---|---|---|---|
| MTG-Jamendo | 56 | 9949 | 3802 | 4231 | 216.4 |
| CAL500 | 17 | 400 | 50 | 50 | 186.5 |
| Emotify | 9 | 320 | 40 | 40 | 59.7 |

Table 1: Overview of the datasets, including train, validation, and test splits.

## 4.2 EXPERIMENTAL SETUP

For our experimental setup, we employ specific train-test-dev splits for each dataset to ensure a standardized evaluation process. For the MTG-Jamendo dataset, we utilize the official split-0 provided by the dataset's authors[2]. For the CAL500 and Emotify datasets, no publicly available splits exist since these datasets were initially designed for evaluation purposes. To address this, we define our own train-test-dev splits, which we plan to release in the future to facilitate reproducibility and benchmarking of our work. The details of these splits are provided in Table 1. For the MERT feature extraction, we split the songs in 10 second segments and we utilise the widely-used MERT-v1-95M model available on Hugging Face[3]. For LLM embeddings, we select Sentence Transformers (Reimers, 2019), given the multi-label nature of all three datasets. Specifically, we use the all-MiniLM-L6-v2 model[4], which has been fine-tuned on 1 billion sentence pairs from a pre-trained MiniLM-L6-H384-uncased model. We leverage the Hugging Face implementation for this approach. The MERT features for each 10s fragment are averaged and fed into a shallow two-layered attention network to project these MERT features to the LLM embedding space. We implement Mean Shift Clustering (Fukunaga & Hostetler, 1975) using the scikit-learn library. The $k$ parameter to obtain the actual labels $\hat{y}$ is set to the average number of labels per instance in the dataset: 2 for Jamendo MTG, 4 for CAL500, and 3 for Emotify. For all the training processes, we ran for 100 epochs with a batch size of 256, using the AdamW optimiser. The base learning rate is fixed to $1e^{-5}$ with a ReduceLROnPlateau scheduler monitoring the validation macro F1 score and a minimum learning rate threshold of $1.6e^{-7}$. The weight decay is set to $1e^{-4}$. All models are trained on 4 NVIDIA Tesla V100 DGXS 32 GB GPUs, and the training and evaluation code is implemented using PyTorch and available online[5].

## 4.3 BASELINES

Our experimental evaluation comprises three distinct phases, each designed to systematically assess the efficacy of our approach for cross-dataset generalization and zero-shot inference. We propose two baselines and finally, an alignment regularization approach:

**Baseline 1 - Single Dataset Training**: The first baseline experiment assesses the model's performance when trained solely on a single dataset. The evaluation is on the test set of the same dataset to gauge in-domain performance, as well as the test sets of the other two datasets to measure cross-dataset generalization. This baseline shows how well the models can generalise across diverse datasets without any external label alignment or regularization.

**Baseline 2 - Clustering of LLM Embeddings**: In the second baseline experiment, we exploit alignment of emotion labels by clustering the label embeddings generated by a Large Language Model across two datasets (see Section 3.2). These clusters capture semantically akin emotions. The model is then trained on this aligned label space, wherein the target emotion is inferred from the

---

[1]URL suppressed for anonymous review

[2]https://github.com/MTG/mtg-jamendo-dataset/tree/master/data/splits/split-0

[3]https://huggingface.co/m-a-p/MERT-v1-95M

[4]https://huggingface.co/sentence-transformers/all-MiniLM-L6-v2

[5]URL suppressed for anonymous review

cluster centroids of the LLM embeddings instead of individual labels. This approach leverages the presence of common emotion clusters, thereby enhancing the model's capacity for generalisation across datasets. In this case, the model is trained on two datasets conjointly and evaluated on the test sets of all three datasets.

**Alignment Regularization**: We build upon the second baseline by incorporating alignment regularization as described in Section 3.4. This regularization technique encourages MERT feature embeddings of semantically similar emotion labels to be mapped to proximate positions within the shared embedding space. This phase aims to refine the model's capacity for generalization to unseen labels and assess the model's zero-shot performance on new dataset with disjoint sets of labels.

**Contrastive Language-Audio Pretraining(CLAP)**: is a pretrained network trained using contrastive learning that puts similar audio embeddings and text embeddings in a shared space compared for non-matching pairs. Researchers have used CLAP to generalize to new tasks and audio categories without task-specific training, relying on the semantic richness and intensive pretraining in CLAP. This encourage us to use CLAP as a baseline to compare zero shot inference performance.

### 4.4 RESULTS

This section details analysis of our results. The evaluation is structured to assess the model's capacity for generalisation from single-dataset training, the enhancement achieved through label clustering utilising LLM embeddings, and the final improvement attained via alignment regularization.

**Baseline 1**: The first set of experiments evaluates the model's performance when trained solely on a single dataset. Table 2 presents the results, showcasing strong in-domain performance with macro F1 scores of 0.120, 0.346 and 0.442 for MTG-Jamendo, CAL500 and Emotify respectively. The large difference in F1 scores between datasets is partly due to a different number of classes: 56, 17, and 9 respectively. The cross-dataset generalization remains a significant challenge, with performance drops of over 50 percent when evaluating on the test sets of other datasets. For example, MTG-Jamendo performance drops from 0.120 to 0.0142 when the training dataset switches to CAL500. The results show that the model's performance declines when evaluated on unseen datasets. This highlights the challenge of training a model solely on a limited set of labels and attempting direct cross-dataset inference without any form of alignment or adaptation. For a better comparison with further baselines, we have also trained models. The full details can be found in Appendix B.

| Trained on \ Tested on | MTG-Jamendo | CAL500 | Emotify |
|---|---|---|---|
| MTG-Jamendo | 0.120 | 0.258 | 0.379 |
| CAL500 | 0.0142 | 0.346 | 0.275 |
| Emotify | 0.0172 | 0.295 | 0.442 |

Table 2: Macro F1 scores for baseline 1 - single dataset training.

**Baseline 2**: In the second baseline, we introduce LLM-based clustering of emotion labels across two datasets. The model is trained on two datasets together, with the emotion labels clustered based on the semantic similarity of their LLM embeddings. Compared to the first baseline, this approach leads to new best individual performance improvements for all datasets. For example, the best macro F1 score on the Emotify dataset increases from 0.4483 to 0.5037 when we combine CAL500 and Emotify through this label alignment technique. We posit this improvement is due to the model's enhanced ability to capture the underlying semantic relationships between emotion labels across the different datasets. Detailed results can be seen in Table 3.

**Alignment regularization**: The final baseline incorporates alignment regularization to further enhance the model's cross-dataset and zero-shot capabilities. By regularizing the MERT feature embeddings of dissimilar examples based on their label embeddings, the model is encouraged to increase the separation between examples of dissimilar emotion labels. This aims to refine the model's ability to distinguish between distinct emotion categories, even when encountering previously unseen labels. Table 4 presents the detailed results for this phase. We observe that the cross-dataset performance is reduced when compared to Baseline 2. For example, CAL500 macro F1 score reduces from 0.350 to 0.332 when trained on CAL500 and Emotify combination (row 2, column 3 in

| Trained on / Tested on | MTG-Jamendo | CAL500 | Emotify |
|---|---|---|---|
| MTG-Jamendo + CAL500 | 0.132 | 0.433 | 0.310 |
| CAL500 + Emotify | 0.0196 | 0.350 | 0.390 |
| Emotify + MTG-Jamendo | 0.108 | 0.319 | 0.341 |

Table 3: Macro F1 scores for baseline 2 - clustering of LLM embedings.

Table 3 vs Table 4). This is because the regularization term accentuates the dissociation of dissimilar emotion labels during training, which in turn diminishes the model's interpolation capacity between known labels. However, the model's zero-shot performance on the new dataset improves significantly, with the F1 score increasing by 15-20% compared to the previous baselines. Specifically, the zero-shot F1 score on the Emotify dataset increased to 0.350 from 0.310 in Baseline 2. Detailed comparisons across different splits are shown in Table 7. These results demonstrate that the alignment regularisation substantially enhances the model's ability to generalise in zero-shot scenarios, where it must effectively handle completely novel emotion labels.

**Comparison with CLAP**: Compared to the CLAP model, our proposed alignment regularization consistently demonstrates superior or equivalent performance across most of the evaluated splits, with notable improvements observed on the Emotify and CAL-500 datasets. While CLAP achieves a slightly better result on the MTG-Jamendo dataset, our method employs a lightweight adapter module that dynamically adapts to the task-specific data distribution, in contrast to the extensive contrastive loss-based pretraining used in CLAP. Particularly in the context of MTG inference, our adapter is trained only on the relatively smaller CAL500 + Emotify datasets (720 songs), compared to CLAP's training on thousands of songs.

| Trained on / Tested on | MTG-Jamendo | CAL500 | Emotify |
|---|---|---|---|
| MTG-Jamendo + CAL500 | 0.0976 | 0.321 | 0.350 |
| CAL500 + Emotify | 0.0214 | 0.332 | 0.457 |
| Emotify + MTG-Jamendo | 0.112 | 0.337 | 0.289 |

Table 4: Macro F1 scores for alignment regularization.

| Phases / Split | Train-MTG+CAL Test-EMO | Train-CAL+EMO Test-MTG | Train-EMO+MTG Test-CAL |
|---|---|---|---|
| Baseline 1 | 0.344 | 0.0153 | 0.292 |
| Baseline 2 | 0.310 | 0.0196 | 0.319 |
| CLAP | 0.246 | **0.032** | 0.240 |
| Alignment Regularisation | **0.350** ($\lambda = 2.5$) | 0.0214 ($\lambda = 1$) | **0.337** ($\lambda = 1$) |

Table 5: Macro F1 scores for zero shot inference. We do not train for evaluation of CLAP, scores are only for zero shot inference. We indicate the empirically determined best $\lambda$ values (Section 3.4) for each split in alignment regularization. **Legend:** MTG: MTG-Jamendo; CAL: CAL500; EMO: Emotify.

**Choice of regularization**: We validate the choice of regularizer in the following way: Instead of regularizing on the pair of examples from disjoint clusters (as described in Section 3.4), we regularize on the positive examples. In that, we tweak $\mathcal{L}_{reg}$ such that in the new formulation, we minimize the distance between elements in the same cluster. i.e., $\mathbf{x}_i, \mathbf{x}_j \in \mathcal{C}_k$. $K'$ is the number of such positive pairs.

$$\mathcal{L}_{\text{reg'}} = \frac{1}{K'} \sum_{(x_i, x_j)} \mathcal{D}(\mathbf{f}_\theta(\mathbf{x}_i), \mathbf{f}_\theta(\mathbf{x}_j)) \tag{10}$$

We carry out small scale experiment. By training on a combination of Jamendo-MTG + CAL500, we observe that the formulation in Section 3.4, i.e., negative pair-based regularization, outperforms

positive pair-based formulation by a considerable margin. Results in Table 6. Based on this, we decided to use negative pair based formulation for all our experiment.

| Regulizer \ Tested on | MTG-Jamendo | CAL500 | Emotify |
|---|---|---|---|
| Positive | 0.0402 | 0.206 | 0.246 |
| Negative | **0.0761** | **0.320** | **0.402** |

Table 6: Comparison of positive and negative formulation of alignment regularization. Train only on MTG-Jamendo+CAL500.

**Discussion**: Our findings show that combining the Jamendo-MTG dataset with others consistently enhanced Jamendo-MTG's performance in Baseline 2. However, this improvement was not observed across the other datasets. For instance, when integrating Emotify with CAL500, Emotify's performance in Baseline 2 declined in comparison to Baseline 1 (0.442 vs 0.387). This can be attributed to CAL500 being a more extensive dataset with greater diversity in data and emotion labels, while Emotify is smaller and focuses on a specific kind of music. Incorporating CAL500 diluted Emotify's performance by introducing music of genres that was less representative of Emotify's concentrated content consisting of a specific genre. Conversely, adding Emotify to Jamendo-MTG introduced focused data that enhanced Jamendo-MTG's performance on that genre.

When merging datasets, we should also be mindful of variations in music emotion datasets and the resulting challenges as outlined by Kang & Herremans (2024). For instance, the added dataset may contain noisier data, due to cloud annotation instead of expert annotation. It may contain different genres of music or fragments of different length. There may also be many labels per instance, versus only one. Our proposed framework can handle data of all these variations, however, the user should be mindful to focus on high-quality data with similar distribution to the desired target music, in order achieve a quality improvement in the model.

Our results suggest that when aiming to improve a dataset's performance, it is advantageous to augment it with another dataset that may be smaller but contains music of the same character present in the original dataset. In such cases, Baseline 2 serves as an effective strategy to enhance the performance of the larger, more diverse dataset.

Furthermore, for tasks requiring zero-shot learning, alignment regularization proves to be more effective. However, when all data and labels are available, Baseline 2 should be preferred, as it tends to yield better results, particularly for larger datasets like Jamendo-MTG, as described earlier.

## 5 CONCLUSION

In this paper, we introduce a novel approach to music emotion recognition by integrating Large Language Model (LLM) embeddings to harmonize label spaces across diverse datasets, and enable zero-shot learning for new emotion categories. Our method not only effectively clusters and aligns emotion labels through LLM embeddings but also employs a novel alignment regularization, enhancing the semantic coherence between music features and emotion labels across disjoint datasets. The experimental results show zero shot performance of 0.402, 0248, 0.262 in terms of macro F1-score for Emotify, MTG-Jamendo, and CAL500 respectively, setting a new benchmark in the field of music emotion recognition. This approach opens up possibilities for broader applications in affective computing where emotional nuances can be universally understood and processed across contextual bounds.

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

# A  APPENDIX

## A.1  INPUT MERT LAYER SELECTION

To identify the most effective layers of MERT for our task, we conducted empirical evaluations using models trained on the MTG-Jamendo dataset and tested them across three datasets based on the segment-level macro F1 score. Following recommendations from the m-a-p team's work on Music Descriptor[6], we initially focused on the 6th layer, which was suggested to be the most effective for emotion identification. Subsequently, we evaluated three configurations: using the 6th layer alone, all layers, and a specific combination of layers (3/6/9/12). Our results demonstrated that the combination of layers (3/6/9/12) provided a balanced performance across both in-distribution (MTG-Jamendo) and out-of-distribution (CAL500 and Emotify) datasets.

| Dataset
Approach | MTG-Jamendo | CAL500 | Emotify |
|---|---|---|---|
| 6th Layer | 8.24 | 28.6 | 38.7 |
| All Layers | 8.37 | 28.6 | **41.3** |
| Ours (3/6/9/12 Layers) | **8.40** | **29.9** | 40.0 |

Table 7: Macro F1 scores for different layer selection strategies on zero-shot inference tasks.

## A.2  FUSED DATASET FOR BASELINE 1

In addition to the Baseline 1 models trained on single datasets (discussed in the main content), we also evaluate models trained on combinations of two out of three datasets (MTG+CAL500, CAL500+Emotify, and Emotify+MTG) to provide a more direct comparison with our clustering and alignment regularization methods. These methods are specifically designed for scenarios involving two-dataset setups.

The results, presented in Table 8, show the segment-level macro F1 scores for these Baseline 1 models, evaluated across MTG-Jamendo, CAL500, and Emotify. For completeness, we include the performance of a model trained on all three datasets (MTG+CAL500+Emotify) as a reference. Training are all datasets yields the best overall performance.

| Evaluation Dataset
Dataset Combination | MTG-Jamendo | CAL500 | Emotify |
|---|---|---|---|
| MTG+CAL500 | 11.6 | **35.4** | 34.4 |
| CAL500+Emotify | 1.53 | 34.8 | 38.7 |
| Emotify+MTG | 10.3 | 29.2 | 40.6 |
| MTG+CAL500+Emotify | **12.0** | 34.7 | **45.0** |

Table 8: Macro F1 scores for Baseline 1 models trained on different dataset combinations.

---

[6]https://huggingface.co/spaces/m-a-p/Music-Descriptor/blob/main/app.py

## A.3 DATASET AND CLUSTERING ANALYSIS

### A.3.1 DATASET INFORMATION

The datasets used in this study include MTG-Jamendo, CAL500, and Emotify. Their basic characteristics are shown in Table 1. To further analyze the emotional labels, we examine overlapping and disjoint tags, as well as the clustering results for each dataset combination.

### A.3.2 LABEL OVERLAPS AND DISJOINT LABELS

Table 9 presents the overlapping and disjoint labels among the datasets. The limited overlap reflects the different annotation schemes across the datasets, with only two shared labels ("happy" and "sad") between MTG and CAL500, while Emotify has no overlaps with either MTG or CAL500. This provides insights into the challenges of aligning labels across datasets.

| Dataset Pair | Overlapped Labels Count | Examples | Disjoint Labels Count |
|---|---|---|---|
| MTG + CAL500 | 2 | happy, sad | 71 |
| CAL500 + Emotify | 0 | null | 26 |
| MTG + Emotify | 0 | null | 65 |

Table 9: Number of overlapping and disjoint labels for each dataset pair.

### A.3.3 CLUSTERING ANALYSIS

We used the mean-shift clustering algorithm to group emotional labels. Unlike k-means, mean-shift does not require predefining the number of clusters (k); instead, it automatically determines the clusters based on density in the feature space. Table 10 summarizes the clustering results for different dataset combinations.

The results show that MTG + CAL500 has fewer clusters compared to other combinations, due to overlapping and semantically similar tags (e.g., "calm, meditative, relaxing"). In contrast, Emotify introduces more unique and less overlapping tags, leading to a higher number of clusters.

Furthermore, we visually demonstrate how different clusters look like in Figure 2 using a graph structure. Nodes with same color belong to the same cluster. Labels (nodes) from the same cluster are shown connected by an edge. Please note, spatial distance here is irrelevant.

| Dataset Pair | Disjoint Labels Count | Number of Clusters |
|---|---|---|
| MTG + CAL500 | 71 | 42 |
| CAL500 + Emotify | 26 | 22 |
| MTG + Emotify | 65 | 62 |

Table 10: Clustering results for different dataset combinations using mean-shift clustering.

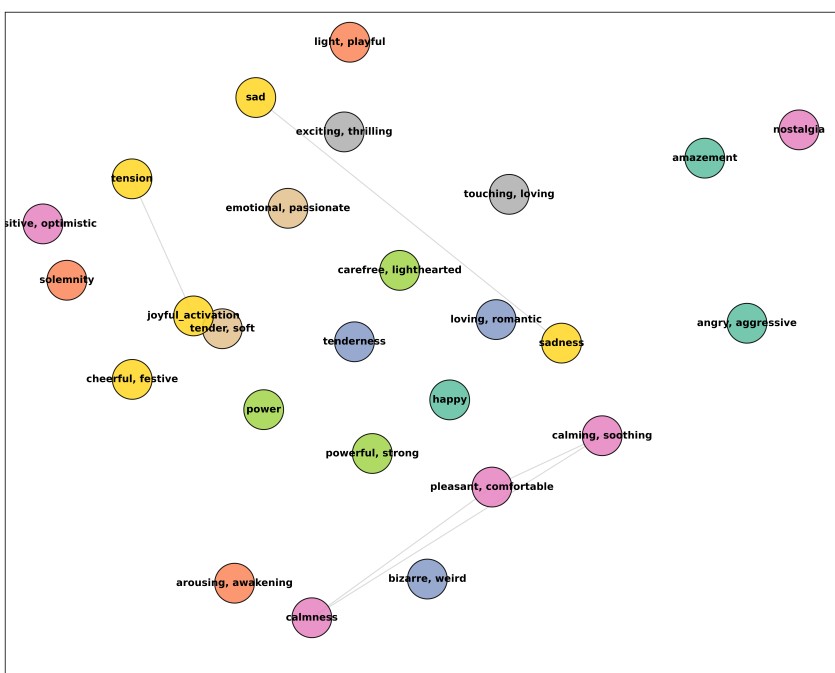

Figure 2: Graph of labels taken from CAL500 and Emotify. Nodes with same color belong to the same cluster. Labels (nodes) from the same cluster are shown connected by an edge. Spatial distance here is irrelevant.

### A.4 Case Study: Bridging Semantic Gaps in Emotional Tags

#### A.4.1 MTG-Jamendo and CAL500 Clustering

In the clustering results for MTG-Jamendo and CAL500, the following emotional labels are clustered together semantically, representing low-energy, soothing emotions:

- MTG-Jamendo: ['calm', 'meditative', 'relaxing', 'soft']
- CAL500: ['calming, soothing', 'pleasant, comfortable']

These labels reflect overlapping interpretations of emotional states across MTG-Jamendo and CAL500 datasets. However, Emotify uses a more concise tag system, where a similar emotional concept is represented by a single label 'calmness'.

Similarly, for a more melancholic or somber emotional tone, MTG-Jamendo and CAL500 uses ['melancholic', 'sad'] while Emotify uses 'sadness'.

#### A.4.2 Bridging the Gap

The challenge lies in reconciling these differences across datasets, where MTG-Jamendo and CAL500 provide a richer set of nuanced labels, while Emotify uses broader, simplified terms. Our proposed method bridges this gap through:

1. **Semantic Alignment:** By using embeddings for emotional labels (e.g., pre-trained language models), our method captures the shared semantic meaning across different label sets. For instance, 'calm, meditative, relaxing' from MTG-Jamendo and 'calming, soothing' from CAL500 are mapped closer to 'calmness' in Emotify, ensuring consistency across datasets without losing interpretative richness.

2. **Clustering-Based Grouping:** Mean-shift clustering automatically groups related emotional labels based on their feature density in the semantic space. This approach allows us to unify nuanced labels like 'pleasant, comfortable' with broader tags like 'calmness', effectively bridging granularity differences.

3. **Zero-Shot Prediction:** Our model leverages shared semantics to generalize across datasets. For example, even if a dataset does not explicitly include a term like 'calming', the model recognizes it as semantically aligned with 'calmness' and makes predictions accordingly.

