# OpenReview forum: "Leveraging LLM Embeddings for Cross Dataset Label Alignment and Zero Shot Music Emotion Prediction"
_ICLR.cc/2025/Conference — Submitted to ICLR 2025_

### Official Review · Reviewer_Kgxv · 2024-10-21

**Soundness:** 3
**Presentation:** 3
**Contribution:** 2
**Rating:** 5
**Confidence:** 3

**Summary:**

The paper proposes a novel approach to music emotion recognition (MER) by utilizing Large Language Model (LLM) embeddings for label alignment across disparate datasets and enabling zero-shot emotion prediction. The core idea involves computing LLM embeddings for emotion labels from different datasets, clustering these embeddings to form a common representation space, and mapping music features derived from the MERT model into this space. The paper introduces an alignment regularization to enhance the model's generalization to unseen data. The proposed method is validated through extensive experiments on three music emotion datasets: MTG-Jamendo, CAL500, and Emotify, demonstrating substantial improvements in zero-shot inference performance.

**Strengths:**

- The use of LLM embeddings for aligning emotion labels across different music datasets is interesting due to its ability to generalize to previously unseen emotion labels
- The experimental results show that the proposed alignment regularization and clustering techniques improve zero-shot performance compared to existing methods

**Weaknesses:**

- The introduction lacks a proper literature review. For example, the claim that "the majority of studies have focused on training and testing with a single dataset" is inaccurate. Since 2022, many audio-language papers (e.g., LTU, CLAP, GAMA) and music-language models (e.g., MuLan, MuLaP) have been trained on multiple emotion recognition datasets.
- The choice of methods, such as the specific LLM used for embeddings, Mean Shift clustering, and the alignment regularization formulation, lacks theoretical support or empirical justification. The paper would benefit from a deeper discussion on why these choices were made and how they compare to alternatives like k-means clustering, l2 regularization, etc.
- The baselines used for comparison are relatively simple. The paper does not include comparisons with multi-task learning, transfer learning techniques, or zero-shot models that could also address cross-dataset generalization. Recent works, for instance, train on valence, arousal, dominance, and other emotion views to avoid dealing with different emotion labels across datasets. Another approach is SSL training across datasets, followed by linear probing on target datasets. For zero-shot approaches, models like CLAP, LTU, and GAMA can support any emotion classes, providing alternative ways to overcome predefined class limitations. Including these baselines would help contextualize the proposed method and offer further analysis on where it outperforms or underperforms existing methods.
- While alignment regularization improves zero-shot performance, it seems to degrade cross-dataset generalization. The paper should provide a more detailed analysis of the trade-offs involved and whether the zero-shot improvement outweighs the generalization loss.

**Questions:**

- It would be beneficial to discuss why previous multi-dataset approaches may fall short and how this work addresses those gaps, expanding on the point mentioned in introduction
- Did the authors have a rationale for selecting specific methods of mean-shift clustering? If so, can the authors add it here and include it in the paper. Adding empirical comparisons, such as how Mean Shift performs relative to k-means or other clustering methods, would help support the hypothesis the authors have
- Can the authors consider adding general baselines that utilize multi-task learning across emotion dimensions, self-supervised learning (SSL) across datasets followed by linear probing, or zero-shot models like CLAP, LTU, and GAMA, which support arbitrary emotion classes? This would provide a clearer picture of the proposed method's strengths and limitations compared to existing techniques and reveal its competitiveness when emotion labels vary across datasets.

---

> ### Author Response · Authors · 2024-11-25
> **clarifications**
>
> Dear reviewer, we thank you for your valuable feedback. We address each point in weakness section as well as the questions you have asked for us.
>
> Weakness:
>
> 1. Audio-language models such as CLAP as well as music language models such as MuLan, are trained on audio(music)-text description pairs. The descriptions may or may not contain any information regarding sentiment, but they are neither explicitly trained using music emotion labels nor are they tested on the downstream task of music emotion recognition. In fact, (following reviewers' suggestion), to the best of our knowledge, we are first to evaluate CLAP's ZSL performance for music emotion recognition task.
>
> 2. Our choice of LLM is driven by popularity of sentence transformers in language modelling community. For clustering, We chose Mean Shift as our clustering method due to its non-parametric nature, which avoids predefining the number of clusters and adapts dynamically to the dataset's inherent properties. This gives us a clear advantage over other popular clustering approach such as k means where k should be predetermined.
>
> 3. Regarding the tradeoff between ZSL and baseline 2, ZSL is a special and extreme scenario when dataset information is not available during training. Based on our observation, if all data and labels are available, Baseline 2 should be preferred, as it tends to yield better results. We have indicated our recommendation in discussion in section 4.4.
>
> Quesitons:
>
> 1. To the best of of knowledge no multi dataset approach exists for music emotion category prediction task. audio-text models such as CLAP also do not evaluate on this downstream task.  To compare, we included LAION CLAP [1] as a reference evaluation. Our results show that our method outperforms CLAP on majority of the zero-shot scenarios. These findings demonstrate effectiveness and superiority of our approach and will be detailed in the revised manuscript.
>
> 2. Mean shift has an inherent advantage over k means due to its non-parametric nature, which avoids predefining the number of clusters and adapts dynamically to the dataset's inherent properties. This ensures flexibility and preserves the semantic diversity of emotion labels. Additionally, Mean Shift is a well-established method known for identifying dense regions in feature space effectively.  The number of clusters automatically recommended by mean-shift are as follows:
>
> |                     | MTG+CAL500 | Emotify+MTG | CAL500+Emotify |
> |---------------------|------------|-------------|----------------|
> | Total No. of Labels | 71         | 65          | 26             |
> | Number of clusters  | 42         | 62          | 22             |
>
> 3. We have added comparision with CLAP in our modified  manuscript. Results show that our method outperforms CLAP on majority of the zero-shot scenarios.
>
> [1] https://arxiv.org/abs/2211.06687

---

> > ### Comment · Reviewer_Kgxv · 2024-11-26
> > **Reviewer response**
> >
> > Appreciate the author's efforts in addressing the questions. However, my concerns regarding the experimental setup remain unresolved. I outline them below:
> > - While the audio-language literature primarily uses audio (music)-text datasets for training, these models are often evaluated on diverse downstream tasks. Reporting zero-shot performance with these models would establish a baseline and should be included in the paper. The authors are encouraged to update the paper with such audio-language model baselines and analysis before the rebuttal PDF update deadline.
> > - If the reviewers compare the proposed method exclusively against zero-shot baselines, it is expected to show improved performance. A more meaningful comparison would involve evaluating the proposed method against alternative setups, such as few-shot learning or other fine-tuning approaches like PEFT or LoRA or linear-probe applied to these baseline models. Notably, since the authors proposed method uses the alignment module shown in Figure 1 which is fine-tuned. Therefore, it becomes critical to compare the proposed method against other fine-tuning techniques from the literature. This would provide stronger evidence that the proposed approach is indeed more effective.
> >
> > I request the authors to address these points to strengthen their evaluation and to make the claims about their proposed method valid.

---

> ### Author Response · Authors · 2024-12-03
>
> Dear reviewer Kgxv,
>
> Thank you for your thoughtful comments. Below, we address your concerns:
>
> 1. While audio-language models like CLAP are often evaluated on various downstream tasks, very few focus on music emotion classification specifically. Most of them evaluate on broader tasks like audio emotion classification. Despite this, we have compared our method (alignment regularisation) with CLAP under two settings:
> - CLAP Zero-Shot: Performance without any fine-tuning on the target dataset.
> - CLAP Linear Probing: Fine-tuning a MLP head on the same dataset's training split.
>
> 2. We compared our approach with CLAP in both zero-shot and linear-probing settings to establish baselines. While advanced fine-tuning techniques like PEFT or LoRA are valid suggestions, they go beyond the scope of our current study, as our method does not involve fine-tuning of the base model (MERT), ensuring a fair comparison.
>
> Here are the results for the three methods. We have append CLAP (zero-shot) result in our appendix and will add the linear probing result in our final version.
>
> |                     | MTG | CAL500| Emotify |
> |---------------------|------------|-------------|----------------|
> | CLAP (Linear Probing) | 0.012         | 0.166          | 0.333             |
> | CLAP (Zero-Shot) | 0.032         | 0.240          | 0.246             |
> | Ours (Zero-Shot) | 0.022         | 0.337          | 0.350             |
>
>
> We hope this clarification addresses your concerns and demonstrates the contributions of our study. We greatly appreciate your consideration of our updated responses as we near the end of the rebuttal period. Thank you again for your time and effort in evaluating our work.
>
> Best Regards

---

### Official Review · Reviewer_PgTs · 2024-11-03

**Soundness:** 3
**Presentation:** 3
**Contribution:** 3
**Rating:** 5
**Confidence:** 4

**Summary:**

The paper presents an approach to use large language model embeddings for label alignments across multiple datasets to perform emotion recognition in music. The paper reports improved performance as a consequence of applying the proposed approach as compared to not using it, and also demonstrated generalization of the proposed approach's efficacy through zero shot learning experiments.

The paper is well motivated, clearly drafted, and provides sufficient background information on prior art. The contributions of the work are clearly outlined. However, there are certain information that are missing which can help to make the work more easily accessible to the reader. Detailed comments and questions are shared below.

**Strengths:**

The article presents an approach to leverage large language model embeddings obtained from labels to generate music emotion clusters for the task of recognizing perceptual emotions from music samples. The proposed approach can be used to perform zero shot learning that can facilitate cross-corpus evaluation. The work is well motivated, the proposed approach is interesting and results show promise.

**Weaknesses:**

The paper lacks certain details which makes it difficult to assess the novelty and the strength of the proposed approach. Some technical contents are missing which can benefit the paper by making it more easily accessible to the reader.

(1) The data description section can benefit from additional details, for example how was the training-validation-test splits obtained?

(2) Providing additional technical details for example how many clusters were used and how the number of clusters was determined will be helpful. Also, it will be informative to share how the performance of the system may vary based on the number of clusters selected? Given that the approach is based on learned emotion clusters from LLM embeddings obtained from the data labels, it will be quite useful for the reader to be aware of some of the decisions that were made while processing the LLM embeddings.

**Questions:**

Here are some detailed questions and comments, which if addressed, can help the paper to be more useful and accessible to the reader:

(1) Section 3.3.1: line 192: "We select the 3rd, 6th, 9th, and12th layers from MERT ..." > what is the rationale behind selecting these specific layers from MERT? If this based on any prior research then providing that reference will be useful. Otherwise, it will be helpful to provide some rationale behind the selection. If the decision was made through empirically studies, please add that information into the paper.


(2) Section 4.1: How were the train-dev-test splits obtained? There are multiple datasets explored in this work, it will be helpful to provide how the data-splits were done and if possible point to some resources that provide such information, that way the results shared in this work can be reproduced.

(4) It is not clear what is used to generate the LLM embeddings. In case of Emotify data, how were the top 3 labels converted to LLM embeddings? It will be helpful if the authors can elaborate more on the steps that were taken to transforming the labels to LLM embeddings and then to emotion clusters. Figure 1 is definitely helpful, but more explanation in section 3 will be helpful.

(5) Section 4.4, page 8, line 408 > a macro-F1 score of 0.078 is fairly low, it will be helpful to explain why the authors specified “strong macro F1 scores”. It will be helpful to provide prior reported results on these datasets to provide some clarity regarding how well the performance shared in this paper perform with respect to prior art.

(6) Section 4.4, page 8, lines 412-414 > Is it possible that these observations indicate that perhaps there may be some over fitting happening during the model training, as a consequence of limited data?

(7) Page 9, table 3: t seems from table 3 that the authors considered clustering across 2-datasets, one rational follow up question would be: what happens when the same is performed across all the three datasets?

---

> ### Author Response · Authors · 2024-11-25
> **clarifications**
>
> Dear reviewer,
> we thank you for your valuable feedback. We address each point in weakness section as well as the questions you have asked for us.
>
> Weakness:
>
> 2. Regarding clustering, we chose Mean Shift as our clustering method due to its non-parametric nature, which avoids predefining the number of clusters and adapts dynamically to the dataset's inherent properties. This ensures flexibility and preserves the semantic diversity of emotion labels. Hence we do not have a say in the numbers of clusters, it is determined and recommended by mean shift algorithm itself based on the LLM embeddings.
>
> Questions:
>
> 1. we initially considered using all layers but found that combining specific layers (3/6/9/12) produced better results in our empirical studies. While prior work, such as the M-A-P Music Descriptor tool [1], identified the 6th layer as optimal for music classification, our results showed that aggregating multiple layers provided a more robust representation. We have appended comparative results demonstrating the improved performance of this approach and included them in Appendix A1 for clarity.
>
> |                        |  MTG  | CAL500 | Emotify |
> |:----------------------:|:-----:|:------:|:-------:|
> | 6th Layer              | 0.082 | 0.286  | 0.387   |
> | All Layers             | 0.083 | 0.286  | 0.413   |
> | Ours (3/6/9/12 Layers) | 0.084 | 0.299  | 0.400   |
>
> 2. We chose Mean Shift as our clustering method due to its non-parametric nature, which avoids predefining the number of clusters and adapts dynamically to the dataset's inherent properties. This ensures flexibility and preserves the semantic diversity of emotion labels. Hence we do not have a say in the numbers of clusters, it is determined and recommended by mean shift algorithm itself based on the LLM embeddings.  As for the exact number of clusters (and the extent of reduction in labels) please see the following table.
>
> |                     | MTG+CAL500 | Emotify+MTG | CAL500+Emotify |
> |---------------------|------------|-------------|----------------|
> | Total No. of Labels | 71         | 65          | 26             |
> | Number of clusters  | 42         | 62          | 22             |
>
> 3. We followed the train-test-validation splits as suggested by the original creators of the  MTG-Jamendo dataset [2]. For the other two datasets, no suggestions are available from the creators. We will publish our splits along with our code for reproducibility. We have added the detailed information in section 4.1 in the revised manuscript.
>
> 4. We used sentence transformer to generate the LLM embeddings, In case of multiple tags. as you have mentioned three in the case of Emotify dataset, we took the average of the three embeddings. These multi-dimensional embeddings are then sent to the clustering algorithm.
>
> 5. Results on MTG-Jamendo has historically been on the lower side. As reported in [3], the highest f1 score reported is 0.209. As for 'strong' f1 scores, we wanted to highlight in-domain performance is strong. For example, f1 score of MTG-Jamendo drops drastically (form 0.07 to 0.02) when training data is changed from MTG-Jamendo to CAL500 in table 2.
>
> 6. Although overfitting might be a contributing factor, we believe the drastic drop in performance is due to data shift and label shift as well. Hence, the application of label embedding clustering, and joint training to target data shift help improve performance in baseline 2.
>
> 7. We acknowledge the importance of exploring fused clustering for comparative purposes and its potential for optimizing overall performance. To address this, we conducted additional experiments for baseline 1 where all three datasets is combined for training. The results show that this approach indeed generates optimal retrieval performance in general. For completeness, we have included these results and a discussion of their implications in Appendix A2 of the revised manuscript.
>
> We hope these clarifications address the reviewer’s concerns. We are committed to improving the manuscript and ensuring it is both useful and accessible to readers. Thank you again for your valuable feedback.
>
> [1] https://huggingface.co/spaces/m-a-p/Music-Descriptor/blob/main/app.py
>
> [2] https://github.com/MTG/mtg-jamendo-dataset/tree/master/data/splits/split-0
>
> [3] https://multimediaeval.github.io/2021-Emotion-and-Theme-Recognition-in-Music-Task/results

---

### Official Review · Reviewer_gHCQ · 2024-11-03

**Soundness:** 2
**Presentation:** 2
**Contribution:** 2
**Rating:** 3
**Confidence:** 5

**Summary:**

The paper introduced a method to conduct zero-shot emotion recognition from music audio. The authors basically followed the well-known zero-shot learning framework. First, label's language embedding is extracted. Second, label embeddings are clustered. Third, audio encoder is learned to match with the clustered label embeddings. Finally, in the inference stage, they conducted zero-shot prediction using unseen label's embedding (or non-parametric clusters). In this work, the authors used LLM embeddings as a labels' language embedding extractor, and MERT audio encoder, Mean Shift clustering, triplet loss and regularization techniques are used. Overall, the paper is easy-to-read. However, the lessons the readers can acquire from the paper is limited in terms of the novelty of the technique, the experimental result, and analysis of the phenomenon.

**Strengths:**

The paper is well-written and easy-to-read. Experimental design seems to be correct and the results seem convincing.

**Weaknesses:**

If we see the result tables (2,3,4), the authors basically trained the model with certain datasets and evaluated the trained model on the dataset that are not used in the training phase. However, I think this is not enough for measuring the zero-shot ability. In MTG-Jamendo, CAL500, and Emotify datasets are distinct in terms of the data instances, however, I think there exist some overlaps in the labels between the datasets. Therefore, what the authors mainly measured through the experiments was instance-level or dataset-level zero-shot prediction capability however, this experimental design may have not considered the zero-shot learning ability on the unseen label thoroughly. Also, the used datasets are mostly multi-labeled emotion datasets, so applying zero-shot learning experimental design that were developed in single-labeled task may not be enough for measuring multi-label zero-shot learning experiments.

**Questions:**

I think if the authors would want to emphasize on the proposed methodology, then the experimental design should be enhanced with more diverse datasets with more strict seen / unseen dataset separation. Also, if the main contribution is on the proposed methodology, then maybe including more tasks and datasets might make the paper more solid, such as zero-shot music tagging, audio tagging, etc. Or, if the main contribution is on zero-shot emotion prediction task itself, then more in-depth experimental design on emotional label might be helpful to highlight the authors contributions. For example, analysis over certain emotion tags that are used in multiple datasets, such as how this emotion tag is used slightly differently over each datasets, and how the proposed method bridge this difference well. Case studies over emotion labels would also be beneficial.

---

> ### Author Response · Authors · 2024-11-25
> **clarifications**
>
> Dear Reviewer,
> Thank you for your detailed feedback and insightful suggestions. Below, we address your concerns regarding the evaluation of zero-shot capabilities and the use of multi-label datasets.
>
>
> 1. Zero-Shot Comparison
>
> We acknowledge your concern about the overlap in labels between the datasets and its potential impact on measuring zero-shot learning. We address this issue as follows:
> Scarcity of Music Emotion Datasets: High-quality music emotion datasets are extremely scarce, limiting the availability of truly disjoint datasets for evaluation. Despite this limitation, we carefully designed our experiments to test the zero-shot generalization of our method across distinct datasets.
> Overlap in Labels: Thank you for raising the point about overlapping labels. Upon thorough investigation, it is found only two overlapping labels—"happy" and "sad"—between MTG-Jamendo and CAL500. In Emotify, the closest labels are "joyful_activation" and "sadness," which, while semantically similar, differ in their word representations. This is precisely where the use of LLM embeddings, such as Sentence-BERT, becomes critical. The semantic embeddings enable the model to handle such nuances effectively, allowing for meaningful zero-shot evaluation.
> We clarify these aspects in the revised manuscript to provided more information about label overlap, cluster information and the challenges of evaluating zero-shot learning in Appendix A3.
>
> 2. Multi-Labeled Emotion Datasets
>
> You are correct that all three datasets used in our study are multi-labeled. This is why we adopted evaluation metrics specifically suited for multi-label tasks:
> F1 Score: We report F1 scores to measure the overall prediction quality across all true labels.
> Top-k Retrieval: To account for the multi-label nature of the datasets, we retrieve the top-k predictions, where k corresponds to the average number of true labels per instance in each dataset (2 for MTG-Jamendo, 3 for CAL500, and 4 for Emotify). This ensures that our evaluation is tailored to the specific characteristics of the datasets.
> We believe these metrics provide a comprehensive assessment of the model's multi-label zero-shot learning capability.
>
> 3. Detailed Case Study
>
> Our focus is indeed on the zero-shot emotion prediction task itself, aiming to address the challenges inherent in predicting music emotions in a zero-shot setting, especially given the scarcity of overlapping labels across datasets.
> We have taken your suggestions into account and enhanced our analysis of emotional labels. Specifically, we have added detailed case study of how similar emotion tags are used across different datasets, highlighting subtle variations and demonstrating how our proposed methodology bridges these differences effectively. These additions are available in our revised manuscript, detailed in Appendix A4.
> We thank the reviewer again for these valuable suggestions, as they significantly enriched the contribution of our work.

---

> ### Author Response · Authors · 2024-12-03
> **Follow-Up on Revisions and Clarifications**
>
> Dear Reviewer gHCQ,
>
> Thank you once again for your thoughtful feedback and for taking the time to review our paper. We have carefully addressed the issues you highlighted and hope our clarifications and revisions have resolved your concerns. If there are any remaining questions or areas requiring further explanation, we would be happy to address them promptly.
>
> We greatly appreciate your consideration of our updated responses as we near the end of the rebuttal period. Thank you for your time and effort in evaluating our work.
>
> Best Regards

---

### Official Review · Reviewer_2E59 · 2024-11-06

**Soundness:** 3
**Presentation:** 2
**Contribution:** 2
**Rating:** 3
**Confidence:** 3

**Summary:**

The authors use sentence embedding (from Sentence-BERT) and MERT to get the embedding of music emotion label and music respectively, and try to align the label and music embeddings through triplet loss and alignment regularization to achieve emotion classification and zero-shot emotion classification. The authors demonstrated the benefit by using sentence embedding, and also the alignment regularization over baseline.

**Strengths:**

Overall, the ablation studies are self-consistent and supportive to the claims. The improvement on the zero-shot recognition capability via alignment regularization is simple yet effective, which can also be applied to other domains.

**Weaknesses:**

Here are my major concerns.

1. The word LLM used here is a bit misleading to me. If I read the "LLM" used in this work properly, it seems to be a sentence transformer that was modified from BERT by using siamese and triplet network for training which can help derive semantically embedding for sentences that can used to calculate cosine-similarity. The LLM here is not the causal large-scale LLM like GPT.
If BERT-style models are non-causal LLM, if the authors want to claim large-scale LLM can generally bring in huge benefits to this problem especially in generalization to unseen labels, I would encourage the authors to break down between causal and non-causal style models.
Especially, some LLMs do not natively output semantic embedding, the authors may want to describe how they do that generally, and also do comparison among different LLMs with different training strategies and model size.

2. The baseline design does not make too much sense to me. From baseline 2 to baseline 1, there are at least two variables, one is the authors use more datasets, and the other is introducing the LLM embedding clustering. Can we have a baseline with LLM that's just based on one single dataset?

3. Do different clustering methods matter? This is not studied and mentioned.

4. The authors described a specific way to extract representations from MERT (e.g., 3/6/9/12 layers). The authors should explain the motivations. As you eventually go to a follow-up linear projection layer, why not use all layers?

5. Overall, the algorithm makes sense but does not show a lot of novelty. One major attracting point is the claim that using embeddings from LLM will help, but the results to support that claim are not strong enough. This also relates to my major concern point 1.

**Questions:**

How zero-shot is achieved is still not clear to me. Will the authors re-do clustering using new unseen labels? Or will the model just map to existing defined centroids?
Equations are not properly labeled, so it also becomes difficult to refer to equations.

---

> ### Author Response · Authors · 2024-11-25
> **clarification**
>
> Dear Reviewer,
> Thank you for your thoughtful feedback and constructive suggestions. Below, we address your concerns and provide clarifications.
>
> 1. Clarification on LLM Terminology
>
> We appreciate your observation regarding the use of "LLM" and its differentiation between causal and non-causal models. In this work, we used Sentence-BERT for text embedding tasks due to its ability to produce semantically meaningful sentence representations. This made it particularly suitable for aligning textual emotion labels with music embeddings. Since our focus is on advancing the music-emotion understanding task using a proven and effective embedding model, we directly take Sentence-BERT as our language encoder. While causal LLMs like GPT offer potential benefits, they do not natively produce semantic embeddings, hence we did not go that avenue. We will clarify this in the revised manuscript.
>
> 2. Baseline Design
>
> We understand the importance of isolating the effects of clustering versus using additional datasets. However, we avoided clustering a single dataset to respect its established annotations provided by the original authors and creators of the datasets. Clustering across multiple datasets harmonizes label semantics, which is a core step in our approach.
> We recognize the gap between using two datasets without clustering and with clustering. To address this, we have appended additional results to demonstrate the impact of clustering more clearly and will include these findings in the revised manuscript. Following table is the result of combining two/three different datasets (first column) and testing on a single dataset. While comparing with baseline 2, we found the results comply with our hypothesis; i.e., clustering improves performance over non-clustering across all combinations of datasets.
>
> |                        | MTG   | CAL500 | Emotify |
> |------------------------|-------|--------|---------|
> | **MTG+CAL500**         | 0.116 | 0.354  | 0.344   |
> | **CAL500+Emotify**     | 0.015 | 0.348  | 0.387   |
> | **Emotify+MTG**        | 0.103 | 0.292  | 0.406   |
> | **MTG+CAL500+Emotify** | 0.120 | 0.347  | 0.450   |
>
> 3. Clustering Methods
>
> We chose Mean Shift as our clustering method due to its non-parametric nature, which avoids predefining the number of clusters and adapts dynamically to the dataset's inherent properties. This ensures flexibility and preserves the semantic diversity of emotion labels. Additionally, Mean Shift is a well-established method known for identifying dense regions in feature space effectively. We will include this rationale in the revised manuscript.
>
> 4. Layer Selection in MERT
>
> Regarding layer selection, we initially considered using all layers but found that combining specific layers (3/6/9/12) produced better results in our empirical studies. While prior work, such as the M-A-P Music Descriptor tool [1], identified the 6th layer as optimal for music classification, our results showed that aggregating multiple layers provided a more robust representation.
> We have appended comparative results demonstrating the improved performance of this approach and included them in Appendix A1 for clarity.
>
> |                        |  MTG  | CAL500 | Emotify |
> |:----------------------:|:-----:|:------:|:-------:|
> | 6th Layer              | 0.082 | 0.286  | 0.387   |
> | All Layers             | 0.083 | 0.286  | 0.413   |
> | Ours (3/6/9/12 Layers) | 0.084 | 0.299 | 0.400   |
>
> 5. Novelty and Strength of Results
>
> To address the concern regarding the strength of our approach, we included LAION CLAP [2] as a reference evaluation. Our results show that our method outperforms CLAP on majority of the zero-shot scenarios. These findings demonstrate effectiveness and superiority of our approach and will be detailed in the revised manuscript.
>
> 6. Zero-Shot Classification
>
> During training, the model is taught to map input music to defined centroids in the embedding space, capturing the semantic relationships between music features and emotion labels. In inference, the model maps input music to a vector in the language space. Candidate tags are also encoded in the same space, and the closest tags are identified using cosine similarity. This approach allows generalization to unseen emotion labels without retraining or re-clustering.
> We will improve the manuscript by providing a clearer explanation of this process and ensure all equations are properly detailed.
>
> [1] https://huggingface.co/spaces/m-a-p/Music-Descriptor/blob/main/app.py
>
> [2] https://arxiv.org/abs/2211.06687

---

> ### Author Response · Authors · 2024-12-03
> **Follow-Up on Revisions and Clarifications**
>
> Dear Reviewer 2E59,
>
> Thank you once again for your thoughtful feedback and for taking the time to review our paper. We have carefully addressed the issues you highlighted and hope our clarifications and revisions have resolved your concerns. If there are any remaining questions or areas requiring further explanation, we would be happy to address them promptly.
>
> We greatly appreciate your consideration of our updated responses as we near the end of the rebuttal period. Thank you for your time and effort in evaluating our work.
>
> Best Regards

---

### Meta-Review · Area_Chair_ZpMq · 2024-12-17

**Metareview:**

The paper introduced a method to conduct zero-shot emotion recognition from music audio across multiple datasets. The authors follow the well-known zero-shot learning framework and extract a language embedding from the label text using SentenceBERT (which they call an "LLM"), then cluster them, and map multiple datasets into the same space. An audio encoder (MERT) is learned to match with the clustered label embeddings and at inference time, zero-shot prediction for unseen labels (using their embedding, or non-parametric clusters) can be used. Mean Shift clustering, triplet loss and further regularization techniques are used to achieve good zero-shot results. Overall, the paper is easy-to-read. However, the lessons the readers can acquire from the paper is limited in terms of the novelty of the technique, the experimental result, and analysis of the phenomenon -- the work may be unique for "music emotion recognition", but does not introduce new datasets or methods.

**Additional Comments On Reviewer Discussion:**

Reviewers al raised largely similar points (e.g. limited novelty, justification/ motivation of the chosen models, some technical clarifications, etc), to which the authors responded, but without changing the reviewers' minds.

For example, authors (in https://openreview.net/forum?id=Gi3SwL98nL&noteId=H88FIaCBlj) clarify that "we are first to evaluate CLAP's ZSL performance for music emotion recognition task" -- this appears to be true, but it's not a convincing argument for acceptance.

One reviewer notes that the authors don't use an "LLM" as commonly understood -- since Sentence-BERT is not a generative model. Authors acknowledge that, and explain why they use Sentence-BERT, but do not otherwise address the issue, and propose a different taxonomy.

---

### Decision · Program_Chairs · 2025-01-22

Reject